# Physics-Informed Neural Network-Based Inverse Framework for Time-Fractional Differential Equations for Rheology

**DOI:** 10.3390/biology14070779

**Published:** 2025-06-27

**Authors:** Sukirt Thakur, Harsa Mitra, Arezoo M. Ardekani

**Affiliations:** School of Mechanical Engineering, Purdue University, West Lafayette, IN 47907, USA; ssukirt@purdue.edu (S.T.); hmitra@purdue.edu (H.M.)

**Keywords:** rheology, anomalous diffusion, fractional modeling, physics-informed machine learning

## Abstract

Modeling complex systems with memory-dependent behavior, such as anomalous diffusion and viscoelasticity, is often hindered by limited data and the computational challenges of inverse problems. Here, we developed a Physics-Informed Neural Network (PINN) framework that incorporates time-fractional derivatives to infer key physical parameters from sparse and noisy data. Despite the recognized potential of fractional models, we found that existing PINN methods largely overlook non-integer dynamics. Using synthetic data and experimental rheology measurements, we showed that our framework could accurately recover parameters such as the generalized diffusion coefficient and fractional orders—even under 25% Gaussian noise. Notably, while traditional models require extensive parameterization, our approach predicted relaxation behavior in biological tissues with fewer assumptions and less than 10% relative error. These results suggest that while fractional PINNs offer powerful tools for learning hidden dynamics, the benefits may be constrained when noise levels are extreme or when behavioral complexity exceeds the model structure.

## 1. Introduction

Fractional calculus broadens the traditional concepts of differentiation and integration to include non-integer orders. Emerging from a question posed by Leibniz in 1695, the field has since matured through the contributions of many mathematicians and now provides a powerful extension of classical calculus. While fractional calculus is often perceived as distinct from integer-order calculus, it essentially serves as an extension, offering a broader and more natural framework. Here, integer-order derivatives emerge as special cases within this framework [1].

The practical relevance of fractional calculus has significantly increased since the 1970s when it gained attention from applied scientists and engineers. This surge in interest stemmed from the recognition that fractional calculus provides enhanced models for many intricate phenomena across various domains. These include electromagnetism [2], rheology [3,4], fluid mechanics [5], and biology [6]. Fractional calculus excels in capturing the memory and non-local effects inherent in these systems, aspects often inadequately described by integer-order derivatives. Fractional models frequently entail numerous parameters, such as fractional orders and coefficients, which are not directly measurable or specifiable [7].

Time-fractional diffusion equations (TFDEs) generalize classical diffusion equations by incorporating derivatives of non-integer order with respect to time, capturing memory effects that are characteristic of anomalous diffusion processes. Unlike classical diffusion, which assumes that the mean square displacement of particles grows linearly with time, TFDEs describe subdiffusive behavior where this growth follows a power law, often observed in heterogeneous or disordered media [8,9]. Mathematically, these equations replace the first-order time derivative with a Caputo or Riemann–Liouville derivative of order α∈(0, 1), allowing for non-local temporal dynamics that integrate the system’s entire history [10,11]. This framework has been successfully applied to model diverse phenomena, including charge transport in amorphous semiconductors [12], diffusion in porous media [13], and subcellular transport in biological systems [14]. The inclusion of fractional time derivatives often leads to new analytical challenges but also enables more accurate descriptions of observed dynamics where traditional models fail [15,16].

Over the years, the study of inverse problems involving fractional-order derivatives has evolved to address the estimation of these parameters from observational data. Early methods relied on traditional optimization and regularization strategies, which faced challenges in terms of stability and computational feasibility. More recent approaches leverage machine learning and data-driven frameworks to improve the robustness and efficiency of these estimations.

Physics-Informed Neural Networks (PINNs) [17] have emerged as powerful computational frameworks, especially for inverse problems. PINNs have been used to solve a vast range of problems in fluid mechanics [18,19], rheology [20,21], and mass transport [22,23]. Despite their broad applicability, most implementations of PINNs primarily address problems involving integer-order derivatives, typically computed via automatic differentiation. However, a growing interest exists in extending PINNs to handle fractional-order derivatives, which offer a more nuanced understanding of complex phenomena. One promising approach involves utilizing the finite difference method to compute fractional-order derivatives, thereby paving the way for the development of fractional-PINNs [24]. This novel methodology has already yielded promising results, facilitating the discovery of fractional-order models for turbulent flow [25] and epidemiological dynamics [26]. However, it is worth noting that the L1 scheme [27], commonly employed to calculate time-fractional derivatives in these contexts, is tailored exclusively for sub-diffusion with derivatives within the range (0, 1), limiting its applicability to capturing only integer-ordered time derivatives.

This study focuses on time-fractional diffusion equations characterizing anomalous diffusion and the fractional Maxwell model governing viscoelastic behavior. To tackle these complex phenomena, we employ a finite difference method [28] tailored for fractional orders within the range [0, 1]. This approach facilitates the calculation of fractional-order derivatives, of which integer-order derivatives are special cases. We aim to integrate this numerical approach into a PINN framework, enabling the simultaneous learning of model parameters and hidden dynamics from limited or noisy data. An overview of fractional derivatives, anomalous diffusion, and fractional viscoelasticity is provided in Section 2.1, Section 2.2, and Section 2.3, respectively. Subsequently, in Section 2.4, we elucidate the methodology devised for solving inverse problems associated with the fractional diffusion equation and the fractional Maxwell model. Notably, our approach ensures that all losses are weighted according to the inherent scales present in the data. Lastly, the outcomes of our investigations into inverse problems are meticulously presented and analyzed in Section 3. Through this structured methodology, we provide a detailed account of how fractional PINNs can be effectively employed to uncover latent parameters and improve understanding of complex physical systems.

## 2. Problem Setup and Methodology

### 2.1. Fractional Derivatives

While offering profound insights into complex phenomena, fractional calculus introduces notable computational and analytical challenges. The non-local characteristics of fractional derivatives render them computationally intensive and more difficult to implement than their integer-order counterparts. These challenges are particularly pronounced in inverse problems, where stability, uniqueness, and computational cost become critical issues.
The Caputo fractional derivative [1] of a function f(t) of order α∈(0, 1) is defined as follows:
(1)$$\frac{\partial^{\alpha }f\left(t\right)}{\partial t^{\alpha }}=\frac{1}{\Gamma (1-\alpha )}\int_{0}^{t}\frac{\partial f(s)}{\partial s}\frac{ds}{{\left(t-s\right)}^{\alpha }},$$ where *s* is the variable of integration and Γ is the gamma function. For α=0, Equation (Equation 1) corresponds to the classical Helmholtz elliptic equation, and for the limit α→1, the integer-order time derivative can be obtained. For time-fractional differential equations, it is possible to numerically approximate the value of the fractional derivative. For 0≤α≤1, the finite difference approximation of the Caputo fractional derivative ($L_{t}^{\alpha }$) of order α can be evaluated as [28](2)$$L_{t}^{\alpha }f\left(t_{k+1}\right):=\frac{1}{\Gamma (2-\alpha )}\sum_{j=0}^{k}b_{j}\frac{f\left(t_{k+1-j}\right)-f\left(t_{k-j}\right)}{\Delta t^{\alpha }},$$
where *k* is the time step number and Δt is the time step size. The coefficients $b_{j}$ are defined as(3)$$b_{j}:={\left(j+1\right)}^{1-\alpha }-j^{1-\alpha }.$$

We can rearrange Equation (Equation 2) to separate the summation into two parts, which simplifies the computation. This results in(4)$$\begin{array}{cc} L_{t}^{\alpha }f\left(t_{k+1}\right):= & \frac{1}{\Delta t^{\alpha }\Gamma \left(2-\alpha \right)}(b_{0}f\left(t_{k+1}\right)-b_{0}f\left(t_{k}\right)\\ \; & +\sum_{j=0}^{k-1}b_{j+1}f\left(t_{k-j}\right)-\sum_{j=1}^{k}b_{j}\;f\left(t_{k-j}\right)). \end{array}$$

### 2.2. Anomalous Diffusion

Diffusion, driven by the erratic thermal motion of molecules, underpins numerous natural phenomena across physical, chemical, and biological realms. In classical Fickian diffusion, transport dynamics align neatly with Gaussian statistics, as evidenced by the linear time evolution of mean squared displacement ($<x^{2}\left(t\right)>$). This linear relationship is captured by Equation [8]:(5)$$<x^{2}\left(t\right)>∼D_{1}t,$$
where $D_{1}$ is the diffusion coefficient illustrating that the average distance traversed by diffusing particles grows proportionally with the square root of time. Here, ∼, denotes asymptotic proportionality [8]. However, in intricate systems like porous media, amorphous semiconductors, and polymeric structures, transport behavior often diverges from this simplistic pattern, heralding what is known as anomalous diffusion. Anomalous diffusion is typified by a nonlinear augmentation of mean squared displacement over time, characterized by the power law expression:(6)$$<x^{2}\left(t\right)>∼\tilde{D}t^{\alpha }.$$

Here, α denotes the anomalous diffusion exponent, and $\tilde{D}$ denotes the generalized diffusion coefficient. The exponent α measures the degree of deviation from normal diffusion. When α=1, the diffusion is normal and follows Fick’s law. When α>1, the diffusion is faster than Fickian diffusion and is called super-diffusion. When α<1, the diffusion is slower than Fickian diffusion and is called sub-diffusion. Typical diffusion processes can be described by the diffusion equation with integer-order derivatives:(7)$$\frac{\partial c}{\partial t}=\nabla \cdot \left(D_{1}\nabla c\right).$$

Here, c denotes the concentration of diffusing particles, and ∇ denotes the gradient operator. Processes with anomalous diffusion are described by fractional-order derivatives. For example, sub-diffusion can be modeled by using a time-fractional derivative of order α, where 0<α<1, as follows:(8)$$\frac{\partial^{\alpha }c}{\partial t^{\alpha }}=\nabla \cdot \left(\tilde{D}\nabla c\right),$$
where $\tilde{D}$ denotes the sub-diffusion coefficient. This equation implies that the diffusing particles have a long-tailed waiting time distribution, meaning they tend to stay longer in some regions before moving to another. The jump length distribution, however, remains finite. In many real-world scenarios, such as transport in porous media or biological tissues, the diffusion coefficient is not constant but depends on the local concentration of the substance. To account for such non-Fickian effects, we consider a concentration-dependent diffusion coefficient within the fractional framework. The generalized time-fractional diffusion equation then becomes(9)$$\frac{\partial^{\alpha }c}{\partial t^{\alpha }}=\tilde{D}\left(c_{xx}+c_{yy}\right)+{\tilde{D}}_{c}\left(c_{x}^{2}+c_{y}^{2}\right),$$
where the subscripts *x*, *y*, and *c* denote the derivatives with respect to the *x*-coordinate, *y*-coordinate, and the concentration, respectively. This form can be derived by applying the chain rule to the diffusion term $\nabla \cdot (\tilde{D}\left(c\right)\nabla c)$ in two spatial dimensions, where $\tilde{D}\left(c\right)$ is a function of concentration. Such models are useful for capturing nonlinear and anomalous transport dynamics [29].

### 2.3. Fractional Viscoelasticity

The theory of linear viscoelasticity offers a robust mathematical framework for understanding the interplay between stress (τ(t)), strain (ϵ(t)), and time (*t*). A common examination method, known as a relaxation test, involves subjecting a material to a constant strain and analyzing its stress response. In such tests, the resulting stress (τ(t)) can be expressed as(10)$$\tau \left(t\right)=G\left(t\right)\epsilon^{0},$$
where G(t) is defined as the relaxation modulus and $\epsilon^{0}$ is the strain applied at the initial time. The traditional linear viscoelastic models are built using springs and dashpots. An ideal spring, obeying Hooke’s law, relates stress and strain as(11)τ(t)=Eϵ(t),
while a dashpot, adhering to Newton’s model, equates stress to the derivative of strain with respect to time:(12)$$\tau \left(t\right)=\eta \frac{d\epsilon (t)}{dt},$$
where E is the elastic modulus and η is the material viscosity. These elements can be interconnected to form more complex models, with the relaxation modulus expressions derived accordingly. Connecting a spring and a dashpot in series yields the Maxwell model, while parallel connection results in the Kelvin–Voigt model. Introducing additional elements facilitates the capture of multiple timescales in material behavior. However, the incorporation of numerous elements poses challenges. Interpretation becomes more intricate, and the model becomes increasingly susceptible to overfitting as the number of elements rises. An alternative approach to capturing complex material behavior with fewer parameters involves the utilization of a viscoelastic element known as the springpot. In the case of a springpot, the stress–strain relationship is defined as(13)$$\tau \left(t\right)=\kappa \frac{d^{\nu }\epsilon \left(t\right)}{dt^{\nu }},$$
where κ is the ‘firmness’ of the material and ν is the order of the fractional derivative. The springpot’s behavior varies with the value of the fractional derivative order. The fractional Maxwell model is obtained by replacing the spring and dashpot of the classic Maxwell model with two springpots connected in series. The stress–strain relationship for a fractional Maxwell model is given by [6](14)$$\tau \left(t\right)+\eta \frac{d^{\mu }\tau \left(t\right)}{dt^{\mu }}=\kappa \frac{d^{\nu }\epsilon \left(t\right)}{dt^{\nu }},$$
with the relaxation modulus being defined as(15)$$G\left(t\right)=\frac{\kappa }{\eta }t^{\mu -\nu }E_{\mu ,1+\mu -\nu }\left(-\frac{t^{\mu }}{\eta }\right),$$
where $E_{\nu }$ is the generalized two-parameter Mittag-Leffler function. The Mittag-Leffler function is defined as [30](16)$$E_{a,b}\left(z\right)=\sum_{n=0}^{\infty }\frac{z^{n}}{\Gamma (an+b)},$$
where Γ is the gamma function. The fractional Maxwell model has the capability of capturing long memory effects and can capture complex material behavior with fewer parameters.

### 2.4. Physics-Informed Neural Networks

While automatic differentiation enables the computation of integer-order derivatives, the direct calculation of fractional-order derivatives presents a challenge. To address this, in computing the residual of a time-fractional differential equation, we employ automatic differentiation for spatial order derivatives and finite differences for time derivatives [24], utilizing Equation (Equation 4).

In our pursuit to infer the anomalous diffusion coefficient as a function of concentration and the fractional-order derivative, we approximate the functions $\left(t,x,y\right)\mapsto \left(c^{pu}\right)$ and $\left(c^{pu}\right)\mapsto \left(D\right)$ using two neural networks parameterized by θ and ϕ, respectively. Here, the superscripts ‘pu’ denote the physics uninformed nature of these networks.

To quantify the loss for regression over the concentration field, we define the mean squared loss as follows:(17)$$L_{data-AD}\left(\theta \right)=E_{\left(t,x,y,c\right)}\left[\frac{|c^{pu}\left(t,x,y;\theta \right)-c{|}^{2}}{{\sigma_{c}}^{2}}\right],$$
where E denotes the expectation approximated by the population mean, $\sigma_{c}$ is the standard deviation of the reference concentration field *c*, and the subscript AD is used to denote anomalous diffusion. We now define(18)$$h\left(c,t^{k}\right)=b_{0}c\left(t^{k}\right)-\sum_{j=0}^{k-1}b_{j+1}c\left(t^{k-j}\right)+\sum_{j=1}^{k}b_{j}c\left(t^{k-j}\right),$$
where $t^{k}$ is the $k^{th}$ time step. Further, let(19)$$g\left(c,D\right)=\tilde{D}\left(c_{xx}+c_{yy}\right)+{\tilde{D}}_{c}\left({\left(c_{x}\right)}^{2}+{\left(c_{y}\right)}^{2}\right),$$
as defined in Equation (Equation 9). To construct the physics-informed network for anomalous diffusion, we combine Equations (Equation 4) and (Equation 9) to get(20)$$\begin{array}{cc} c^{pi}\left(t^{k+1},x;\Delta t,\theta ,\phi ,\alpha \right)= & \Gamma \left(2-\alpha \right)\Delta t^{\alpha }g\left(c^{pu}\left(t^{k},x;\theta \right),\tilde{D}\left(c^{pu};\phi \right)\right)\\ \; & +h(c^{pu}\left(t^{k},x;\theta \right),t^{k}), \end{array}$$
where x=(x,y) are the spatial coordinates and the superscript ‘pi’ is used to denote ‘physics-informed’. Since the physics-uninformed and physics-informed concentration fields evaluated at the same spatio-temporal point need to be consistent, this allows us to define a consistency loss [31] as(21)$$L_{consistency-AD}\left(\theta ,\phi ;\Delta t,\alpha \right)=E_{\left(t,x\right)}\left[\frac{|c^{pi}\left(t,x;\Delta t,\theta ,\phi ,\alpha \right)-c^{pu}(t,x;\theta ){|}^{2}}{{\sigma_{c}}^{2}}\right].$$

The parameters θ and ϕ can now be optimized by minimizing the following mean squared error function:(22)$$L_{MSE-AD}\left(\theta ,\phi ;\alpha \right)=L_{data-AD}\left(\theta \right)+L_{consistency-AD}\left(\theta ,\phi ;\alpha \right).$$

Similarly, to learn the parameters of the fractional Maxwell equation (Equation (Equation 14)), we utilize data on stress (τ) and strain (ϵ) as functions of time (*t*). We approximate the functions $\left(t\right)\mapsto \left(\sigma^{pu}\right)$ and $\left(t\right)\mapsto \left(\epsilon^{pu}\right)$ using two fully-connected neural networks parameterized by β and ζ, respectively. The mean squared loss for regression over the stress and strain fields is defined as(23)$$L_{data-FM}\left(\beta ,\zeta \right)=E_{\left(t,\tau \right)}\left[\frac{|\tau^{pu}\left(t;\beta \right)-\tau {|}^{2}}{{\sigma_{\tau }}^{2}}\right]+E_{\left(t,\epsilon \right)}\left[\frac{|\epsilon^{pu}\left(t;\zeta \right)-\epsilon {|}^{2}}{{\sigma_{\epsilon }}^{2}}\right],$$
where $\sigma_{\tau }$ and $\sigma_{\epsilon }$ are the standard deviations of the reference stress field τ and the reference strain field, respectively. To get the physics-informed stress, we combine Equations (Equation 4) and (Equation 14) to derive(24)$$\tau^{pi}\left(t^{k+1};\Delta t,\beta ,\zeta ,\nu ,\mu \right)=\kappa L_{t}^{\nu }\left(\epsilon^{pu}\left(t^{k+1};\Delta t,\zeta \right)\right)-\eta L_{t}^{\mu }\left(\tau^{pu}\left(t^{k+1};\Delta t,\beta \right)\right),$$
which allows us to define(25)$$L_{consistency-FM}\left(\beta ,\zeta ;\Delta t,\nu ,\mu \right)=E_{\left(t\right)}\left[\frac{|\tau^{pi}\left(t;\Delta t,\beta ,\zeta ,\nu ,\mu \right)-\tau^{pu}\left(t;\beta \right){|}^{2}}{{\sigma_{\tau }}^{2}}\right],$$
where the subscript FM is used to denote fractional Maxwell. The parameters β and ζ are then optimized by minimizing the following mean squared error function:(26)$$L_{MSE-FM}\left(\beta ,\zeta \right)=L_{data-FM}\left(\beta ,\zeta \right)+L_{consistency-FM}\left(\beta ,\zeta ;\nu ,\mu \right).$$

The PINNs setup for anomalous diffusion and the fractional Maxwell model are shown in Figure 1.

## 3. Results and Discussion

### 3.1. Numerical Dataset—Anomalous Diffusion

We constructed a synthetic dataset to serve as a benchmark and evaluate the efficacy of our framework. Our neural network architecture comprised a fully connected network with 4 layers and 16 neurons per layer to approximate the function mapping $\left(t,x,y\right)\to \left(c^{pu}\right)$, and another fully connected network with 2 layers and 4 neurons per layer to approximate the subsequent mapping $\left(c^{pu}\right)\to \left(D\right)$. Throughout this study, we employed the swish activation function for all neural networks.

In our training process, we adopted a cosine learning rate schedule [32]. Specifically, we set the maximum learning rate ($\eta_{max}$) to $2.5\;\times \;10^{-3}$ and the minimum learning rate ($\eta_{min}$) to $2.5\;\times \;10^{-6}$, enabling us to compute the learning rate (η) dynamically using the following formula:$$A=A_{min}+0.5\left(A_{max}-A_{min}\right)\left(1+\cos\left(\frac{T_{cur}}{T_{max}}\pi \right)\right),$$
where $A_{min}$ and $A_{max}$ are the minimum and maximum values of the learning rate, respectively, $T_{cur}$ denotes the current time step, and $T_{max}$ represents the total number of time steps. These values were selected based on prior PINN studies and preliminary experiments. A higher initial learning rate ($2.5\;\times \;10^{-3}$) promotes fast exploration, while a smaller final rate ($2.5\;\times \;10^{-6}$) supports stable convergence. Cosine annealing effectively balances this trade-off in non-convex PINN optimization. The neural networks underwent training for 100,000 iterations and the optimized parameters were obtained by minimizing the loss in Equation (Equation 22).

To assess the robustness of our framework, we introduced Gaussian noise to the concentration field and analyzed its impact on the predicted generalized diffusion coefficient and fractional derivative order. Figure 2a compares the reference generalized diffusion coefficient and its predicted counterparts across different noise levels as a function of concentration. While the predictions closely follow the true values, minor deviations appear as noise increases, particularly at higher concentrations.

Figure 2b shows the relative errors in the predicted generalized diffusion coefficient for various fractional orders (α). The errors remain consistently low across all noise levels, demonstrating the robustness of our solver. While the predictions are accurate across all values of α, higher noise levels introduce slight uncertainty, particularly for lower fractional orders.

Figure 2c illustrates the relative errors in the predicted fractional derivative order. While the errors remain small overall, a noticeable trend emerges where lower fractional orders (e.g., α=0.3) exhibit higher sensitivity to noise, as evidenced by more prominent fluctuations in the relative error. Predictions for higher values of α remain comparatively stable, showing limited variation even at the highest noise levels. The corresponding predicted values at different noise levels are reported in Table 1.

These results demonstrate that our solver effectively captures both the generalized diffusion coefficient and fractional derivative order across all tested scenarios. These findings are consistent with previous research suggesting that Physics-Informed Neural Networks (PINNs) exhibit strong resilience to Gaussian noise [21,31]. This robustness is largely attributed to the inclusion of physical constraints during training, which act as implicit regularizers and mitigate overfitting to noisy data. Our observations reaffirm this behavior, indicating reliable performance even under moderate noise conditions. While the introduction of noise slightly affects accuracy, the framework remains robust and reliable under different levels of added noise.

### 3.2. Experimental Dataset—Fractional Maxwell Model

To evaluate our framework’s performance on experimental data to learn the parameters of the fractional Maxwell model, we leverage the nonlinear stress–strain and corresponding relaxation modulus data obtained from shear rheology experiments conducted on minipig tissues by Mitra et al [33]. We plot a sample of the experimentally obtained stress and strain curves in Figure 3. Our neural network architecture consists of a fully connected network with 2 layers and 20 neurons per layer, tasked with approximating the function mapping $\left(t\right)\to \left(\sigma^{pu}\right)$. Additionally, another fully connected network with the same architecture is employed to approximate the subsequent mapping $\left(t\right)\to \left(\epsilon^{pu}\right)$.

We iteratively learn the parameters of Equation (Equation 14) by minimizing the loss function defined in Equation (Equation 26). Subsequently, we utilize these learned parameters to predict the relaxation modulus as defined in Equation (Equation 15). Our study encompasses three distinct tissue samples: neck, belly, and breast tissue. Figure 4 demonstrates a strong agreement between the predicted and experimental relaxation moduli across all three tissue types—neck, belly, and breast—throughout the entire temporal domain. The predicted curves closely follow the experimental trends, accurately capturing the decay behavior of the relaxation modulus over time. This alignment highlights the capability of our framework to generalize across different tissue types with varying viscoelastic properties. Quantitatively, the relative errors between predicted and experimental values remain consistently below 10% (Table 2) for all three cases, underscoring the robustness and accuracy of the proposed method.

## 4. Conclusions and Future Work

Time-fractional differential equations find widespread applications across various fields, albeit often posing computational challenges and instability issues, especially in the context of inverse problems. Recognizing this, there is growing interest in leveraging Physics-Informed Neural Network-based frameworks to address inverse problems entailing time-fractional derivatives. In such frameworks, the fractional time derivative can be efficiently computed using finite differences while automatic differentiation handles other derivatives.

In this study, we tackle two inverse problems associated with (1) anomalous diffusion and (2) fractional viscoelasticity employing the aforementioned approach. Our methodology involves defining the residual loss in a manner that facilitates scaling the loss terms with the standard deviation of the observed data. We utilize numerically generated datasets and experimental data for learning the fractional coefficient and the concentration-dependent generalized diffusion coefficient, as well as for calibrating parameters for the fractional Maxwell model using stress and strain data over time.

To evaluate the performance of our framework in handling anomalous diffusion, we conducted rigorous testing by comparing our predictions for a concentration-dependent generalized diffusion coefficient and the fractional order of the derivative against synthetically generated reference values. Remarkably, even with the addition of 25% Gaussian noise to the concentration dataset, our framework demonstrated robust performance. Specifically, we observed that the relative error in predicting the generalized diffusion coefficient and the order of the fractional derivative was less than 10%. This outcome underscores the resilience and accuracy of our framework, even in the presence of significant noise, reaffirming its reliability in practical applications.

To validate our findings, we predict relaxation moduli for three distinct samples of pig and human subcutaneous tissues from the belly region and compare them against reported values in the literature. The relative errors are consistently below 10%, highlighting the efficacy of the fractional model, which requires fewer parameters. Furthermore, our approach holds promise for extending to model nonlinear fractional viscoelasticity and can readily incorporate experimental data to validate anomalous diffusion predictions. Additionally, the solver’s applicability can be extended to three-dimensional scenarios, offering a broader scope for future exploration into equations involving time-fractional derivatives.

## Figures and Tables

**Figure 1 biology-14-00779-f001:**
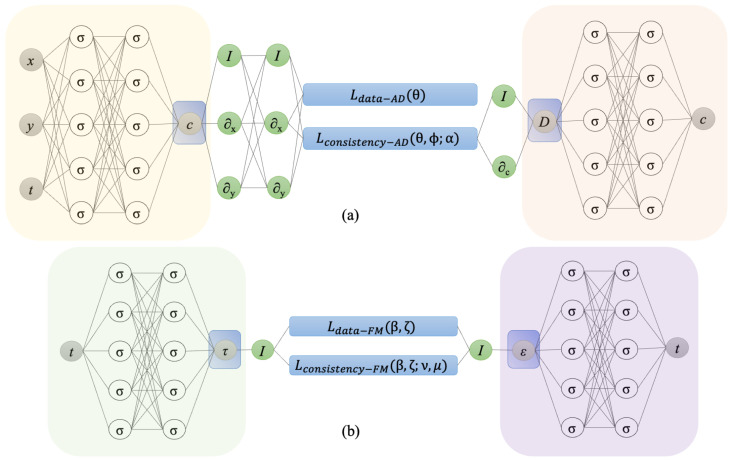
The problem setup for (**a**) anomalous diffusion and (**b**) the fractional Maxwell model, as described in Section 2.4. Here, I denotes the identity operator, and the differential operators ∂x, ∂y, and ∂c are computed using automatic differentiation. In part (**a**), θ and ϕ represent the parameters of the neural network that predict the concentration (*c*) and the diffusion coefficient (*D*), respectively. The parameter α denotes the order of the time-fractional derivative. In part (**b**), β and ζ represent the parameters of the neural network that predict the stress (τ) and the strain (ϵ), respectively, with ν denoting the order of the time-fractional derivative.

**Figure 2 biology-14-00779-f002:**
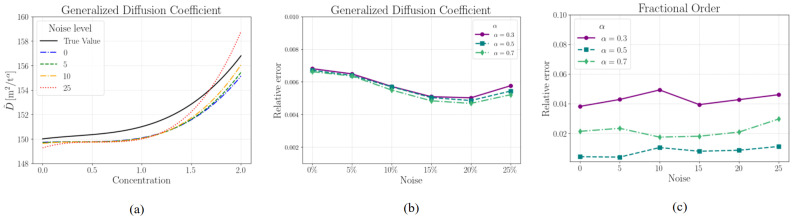
(**a**) The reference generalized diffusion coefficient at different values of concentration compared to different levels of added noise α=0.5. The relative errors in the (**b**) generalized diffusion coefficient and (**c**) fractional order for different levels of added noise. The errors in the generalized diffusion coefficient are comparable across different fractional orders.

**Figure 3 biology-14-00779-f003:**
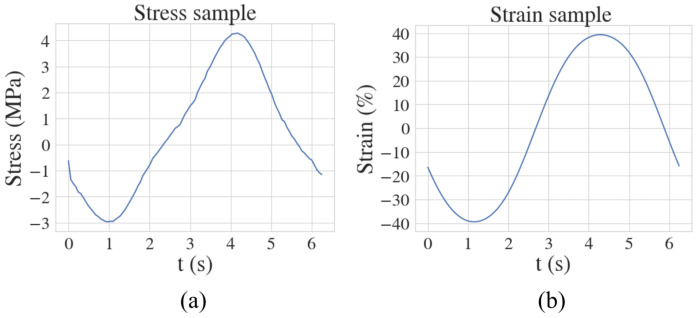
Plot of the experimentally obtained (**a**) stress and (**b**) strain values for a sample reported in [33].

**Figure 4 biology-14-00779-f004:**
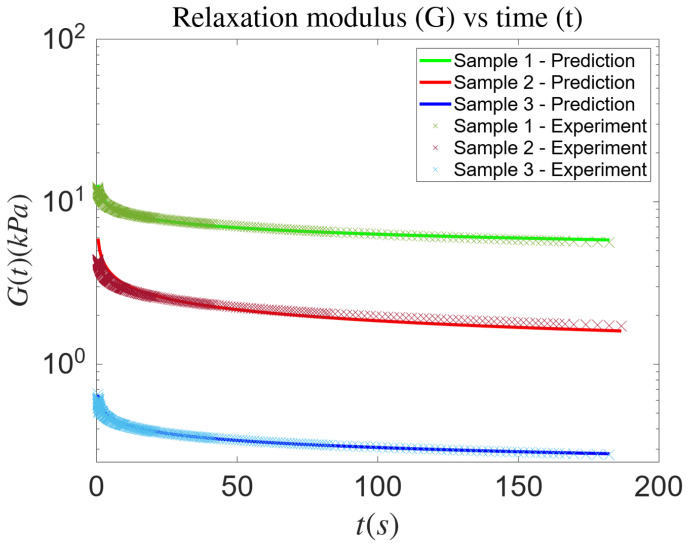
The predicted relaxation modulus and the experimental data corresponding to three samples of minipig skin tissue. Samples 1, 2, and 3 are from the neck, belly, and breast tissue, respectively.

**Table 1 biology-14-00779-t001:** Predicted fractional derivative orders under varying levels of added noise.

True α	0% Noise	5% Noise	10% Noise	15% Noise	20% Noise	25% Noise
0.3	0.311	0.313	0.315	0.312	0.313	0.314
0.5	0.502	0.502	0.505	0.504	0.504	0.506
0.7	0.685	0.683	0.688	0.687	0.685	0.679

**Table 2 biology-14-00779-t002:** The relative errors for the relaxation moduli (G(t)) for the three pig tissue samples.

Samples	Sample 1	Sample 2	Sample 3
Relative error	$7.75\times 10^{-2}$	$6.03\times 10^{-2}$	$5.77\times 10^{-2}$

## Data Availability

The data presented in this study are available upon request from the corresponding author.

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
