# Peer review of "Physics-Informed Neural Network-Based Inverse Framework for Time-Fractional Differential Equations for Rheology"

_biology, 2025, doi:10.3390/biology14070779_

Round 1

Reviewer 1 Report

Comments and Suggestions for Authors

General comments: This study presents a study of the application of PINNs to address inverse problems involving time-fractional derivatives. Original contributions are not clearly highlighted, and numerous problems are noticed throughout the text body. More experiments should be conducted. I feel that this paper has not carefully been prepared and needs significantly to be polished. My conclusion is major revision before considering the publication.

Suggestions:

  1. Theabstract and introduction are needed to rewrite and becomelogical. For example, in paragraphs 1, introduces the history of fractional order, but it is widely known and can be summarized. And in line 39, ‘Addressing inverse problems involving fractional-order derivatives presents…’, you should describe the development process of inverse problems involving fractional-order derivatives, rather than summarizing them.
  2. Introduction, the background of time-fractional derivative diffusion equation is necessary to be abundantandsupported by varies references.
  3. (Page 2 Line 75) Some redundant statements. I don't know the purpose of these two sentences: “Moreover, navigating the realm of…” and “Thus, a robust mathematical foundation…”. They have little connection to the article.
  4. (Page 3 Line 81) Itisthe definition of Caputo fractional derivative, not equation.
  5. (Page 4 Line 116) This equation is not a commonly used time-fractional derivative diffusion equation, you need to carefully consider whether to apply your method to this equation.
  6. (Page 5) Figure2(a), is this the concentration measured at a certain point for this numerical experiment? And I don’t know the You should describe it clearly.
  7. (Page 5) Figure 2(b), there is no explanation why the error decreases as the noise level increases, it's relatively uncommon. More experiments with added noise should be conducted and you should explain it.
  8. (Page 8 Line 206) There are too few numerical experiments. On the one hand, more experiments with added noise should be conducted. On the other hand, consider using real experimental data for inversion.

Author Response

General comments: This study presents a study of the application of PINNs to address inverse problems involving time-fractional derivatives. Original contributions are not clearly highlighted, and numerous problems are noticed throughout the text body. More experiments should be conducted. I feel that this paper has not carefully been prepared and needs significantly to be polished. My conclusion is major revision before considering the publication.

Suggestions:

1. The abstract and introduction are needed to rewrite and becomelogical. For example, in paragraphs 1, introduces the history of fractional order, but it is widely known and can be summarized. And in line 39, ‘Addressing inverse problems involving fractional-order derivatives presents…’, you should describe the development process of inverse problems involving fractional-order derivatives, rather than summarizing them.

Ans.: We thank the reviewer for the comment.  The abstract has now been rewritten and the changes in the introduction are highlighted in red. 

2. Introduction, the background of time-fractional derivative diffusion equation is necessary to be abundantandsupported by varies references.

Ans.: We thank the reviewer for the suggestion.  A background for the time-fractional diffusion equations has been added in the introduction. 

3. (Page 2 Line 75) Some redundant statements. I don't know the purpose of these two sentences: “Moreover, navigating the realm of…” and “Thus, a robust mathematical foundation…”. They have little connection to the article.

Ans.: We thank the reviewer for the comment. The redundant statements have been removed and the paragraph has been rewritten, with the changes marked in red.
“While offering profound insights into complex phenomena, fractional calculus introduces notable computational and analytical challenges. The non-local characteristics of fractional derivatives render them computationally intensive and more difficult to implement than their integer-order counterparts. These challenges are particularly pronounced in inverse problems, where stability, uniqueness, and computational cost become critical issues.”

4. (Page 3 Line 81) Itisthe definition of Caputo fractional derivative, not equation.

Ans.: We thank the reviewer for the comment. We have added the following text to the manuscript:

“The Caputo fractional derivative [1] of a function f(t) of order \alpha in (0, 1) is defined as follows”

5. (Page 4 Line 116) This equation is not a commonly used time-fractional derivative diffusion equation, you need to carefully consider whether to apply your method to this equation.

Ans.: We thank the reviewer for the suggestion. We have added the following text to the manuscript:

“In many real-world scenarios, such as transport in porous media or biological tissues, the diffusion coefficient is not constant but depends on the local concentration of the substance. To account for such non-Fickian effects, we consider a concentration-dependent diffusion coefficient within the fractional framework. The generalized time-fractional diffusion equation then becomes:”

“This form can be derived by applying the chain rule to the diffusion term in two spatial dimensions, where D(c) is a function of concentration. Such models are useful for capturing nonlinear and anomalous transport dynamics.”

6. (Page 5) Figure2(a), is this the concentration measured at a certain point for this numerical experiment? And I don’t know the You should describe it clearly.

Ans.: We thank the reviewer for the question.We have added the following text to the manuscript

“ Fig. 2 (a) compares the reference generalized diffusion coefficient at different values of concentration with the predictions under different noise levels.”

7. (Page 5) Figure 2(b), there is no explanation why the error decreases as the noise level increases, it's relatively uncommon. More experiments with added noise should be conducted and you should explain it.

Ans.: We thank the reviewer for the suggestion. 

We have added the following text to the manuscript

“These findings are consistent with prior studies indicating that Physics-Informed Neural Networks exhibit strong resilience to Gaussian noise [12, 26]. This robustness is often attributed to the incorporation of physical constraints into the learning process, which acts as a regularizer and reduces overfitting to noisy data. Our results reinforce this behavior, showing that even under moderate noise, the predictions remain accurate and stable.”

8. (Page 8 Line 206) There are too few numerical experiments. On the one hand, more experiments with added noise should be conducted. On the other hand, consider using real experimental data for inversion.

Ans.: We thank the reviewer for the suggestion. We have conducted more numerical experiments, adding noise levels 15% and 20% to figured 2 (b) and 2(c). 

We have used real experimental data for inversion for fractional viscoelasticity, and the results are discussed in section 3.2. Experimental dataset - fractional Maxwell model 

Reviewer 2 Report

Comments and Suggestions for Authors

1.In Abstract, "....25% Gaussian noise....", I couldn't find it in the body of the paper. Please provide a detailed description of it in the main text. 
2.In Abstract," Notably, the relative error in predicting the generalized 15
diffusion coefficient and the order of the fractional derivative is less than 10% for all cases,",which is not fully discussed or researched in the main body of the paper, lines 236 to 237. And the  raw data needs to be presented in order for readers to understand the calculation process of the data.
3.Lines 197-199,"...Specifically, we set the maximum learning rate (ηmax) to 2.5e-03 and the minimum learning rate (ηmin) to 2.5e-06, ...",The author set  number , 2.5e-03 and 2.5e-06, of  ηmax and ηmin. What is the basis for setting these data?
4.Fig.4, The prediction and experiment data lines are shown , but legend , in the upper right corner ,is inconsistent with the curve and cannot be distinguished.

Author Response

1.In Abstract, "....25% Gaussian noise....", I couldn't find it in the body of the paper. Please provide a detailed description of it in the main text. 

The authors would like to point out that the effects of varying gaussian noise, including the 25% case has been presented in Fig. 2. The authors have also explained this through the following text in Sec 2.4 as follows,

“To assess the robustness of our framework, we introduced Gaussian noise to the concentration field and analyzed its impact on the predicted generalized diffusion coefficient and fractional derivative order. Fig. 2 (a) compares the reference generalized diffusion coefficient at different values of concentration with the predictions under different noise levels. While the predictions closely follow the true values, minor deviations appear as noise increases, particularly at higher concentrations. Fig. 2b presents the relative errors in the predicted generalized diffusion coefficient for different fractional orders (α). The errors remain relatively low across all noise levels, indicating that our solver maintains accuracy even with increased noise. However, slight variations suggest that higher noise levels introduce some uncertainty, particularly for lower fractional orders. Fig. 2c illustrates the relative errors in the predicted fractional derivative order. While the errors remain small overall, trends vary depending on the value of α, with some configurations exhibiting increased errors at certain noise levels. Notably, the predictions for α = 0.3 show more fluctuation compared to higher values of α. These results demonstrate that our solver effectively captures both the generalized diffusion coefficient and fractional derivative order across all tested scenarios.”

The authors have also added the following text to discuss the noise effects further as following,

“These findings are consistent with prior studies indicating that Physics-Informed Neural Networks (PINNs) exhibit strong resilience to Gaussian noise [20,33]. This robustness is often attributed to the incorporation of physical constraints into the learning process, which acts as a regularizer and reduces overfitting to noisy data. Our results reinforce this behavior, showing that even under moderate noise, the predictions remain accurate and stable”.

2.In Abstract," Notably, the relative error in predicting the generalized 15
diffusion coefficient and the order of the fractional derivative is less than 10% for all cases,",which is not fully discussed or researched in the main body of the paper, lines 236 to 237. And the  raw data needs to be presented in order for readers to understand the calculation process of the data.

The authors thank the reviewer for the comment. Lines 236 to 237 have been updated and the following text has been added to the manuscript

“Figure 4 demonstrates a strong agreement between the predicted and experimental relaxation moduli across all three tissue types—neck, belly, and breast—throughout the entire temporal domain. The predicted curves closely follow the experimental trends, accurately capturing the decay behavior of the relaxation modulus over time. This alignment highlights the capability of our framework to generalize across different tissue types with varying viscoelastic properties. Quantitatively, the relative errors between predicted and experimental values remain consistently below 10% (Table 1}) for all three cases, underscoring the robustness and accuracy of the proposed method.“

Furthermore, the plot of the generalized diffusion coefficient along with the relative errors are shared here

We report the values of the fractional order in Table 1 as well

3.Lines 197-199,"...Specifically, we set the maximum learning rate (ηmax) to 2.5e-03 and the minimum learning rate (ηmin) to 2.5e-06, ...",The author set number , 2.5e-03 and 2.5e-06, of  ηmax and ηmin. What is the basis for setting these data?

The authors thank the reviewer for the question. We have added the following text in the manuscript.

“These values were selected based on prior PINN studies and preliminary experiments. A higher initial learning rate (2.5e-03) promotes fast exploration, while a smaller final rate (2.5e-06) supports stable convergence. Cosine annealing effectively balances this trade-off in non-convex PINN optimization.”

4.Fig.4, The prediction and experiment data lines are shown , but legend , in the upper right corner ,is inconsistent with the curve and cannot be distinguished.

The authors would like to thank the reviewer for pointing this out. We have modified the legend.

Reviewer 3 Report

Comments and Suggestions for Authors

The authors trained two PINNs for two inverses problems modeled by time-fractional differential equations. The manuscript can be accepted if the following comments have been addressed. 

  1. line 95: Define the relation ~
  2. I am not sure whether the journal requires each image to appear on a separate page. If not, the authors may need to adjust the placement of their images to align with the relevant text.
  3. Fix the email address "e-mail@e-mail.com" on the first page. 

Author Response

The authors trained two PINNs for two inverses problems modeled by time-fractional differential equations. The manuscript can be accepted if the following comments have been addressed. 

1. line 95: Define the relation ~

The authors would like to thank the reviewer for the suggestion. We have defined the relationship now by adding the following, “Here, ∼, denotes asymptotic proportionality.”

2. I am not sure whether the journal requires each image to appear on a separate page. If not, the authors may need to adjust the placement of their images to align with the relevant text.

The authors have aligned the images within their respective sections.

3. Fix the email address "e-mail@e-mail.com" on the first page. 

The authors have modified the email.

Round 2

Reviewer 1 Report

Comments and Suggestions for Authors

This version is Ok for publication.